# NOISE-CONTRASTIVE VARIATIONAL INFORMATION BOTTLENECK NETWORKS

## ABSTRACT

While deep neural networks for classification have shown impressive predictive performance, e.g. in image classification, they generally tend to be overconfident. We start from the observation that popular methods for reducing overconfidence by regularizing the distribution of outputs or intermediate variables achieve better calibration by sacrificing the separability of correct and incorrect predictions, another important facet of uncertainty estimation. To circumvent this, we propose a novel method that builds upon the distributional alignment of the variational information bottleneck and encourages assigning lower confidence to samples from the latent prior. Our experiments show that this simultaneously improves prediction accuracy and calibration compared to a multitude of output regularization methods without impacting the uncertainty-based separability in multiple classification settings, including under distributional shift.

## 1    INTRODUCTION

Deep neural networks (DNNs) have become the standard tool for challenging classification tasks, e.g. image classification or semantic segmentation, due to their excellent predictive accuracy. However, predictions of DNNs often tend to be overconfident, leading to miscalibration (Guo et al., 2017). This problem is amplified in the presence of distributional shift in the test data, such as from image corruptions (Ovadia et al., 2019). Multiple methods to regularize the output distribution of DNNs during training have been proposed (Joo & Chung, 2020; Joo et al., 2020; Pereyra et al., 2017; Szegedy et al., 2016) to obtain well-calibrated models. We note, however, that evaluating in-domain uncertainty quantification in terms of model calibration is not sufficient, as it does not indicate how well correct and incorrect predictions can be discerned based on the predictive uncertainty (Ding et al., 2020). As Fig. 1 shows, methods that *indiscriminately* regularize the confidence to improve calibration perform significantly worse at separating correct from incorrect predictions.

To address this, we turn to deep variational information bottleneck networks (VIBN; Alemi et al., 2017). Similarly to output regularization methods, they benefit generalization and calibration (Alemi et al., 2018) but, as we show

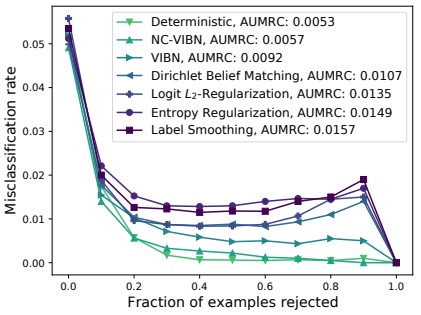

Figure 1: *Ability to separate correct and incorrect predictions based on the predictive entropy* on CIFAR-10: misclassification rate under rejection of uncertain high-entropy examples and the area under the resulting curve. Models trained with explicit output distribution regularization perform substantially worse than a deterministic network. Plain variational information bottleneck networks (VIBN) show a similar tendency.

empirically, suffer from the same separability problem. However, unlike these methods, VIBNs allow us to utilize the distribution matching of the variational approximation to define a noise-contrastive loss term, which can overcome the problem of insufficient separation of correct and incorrect predictions based on the uncertainty while retaining the benefits of the VIBN in terms of generalization and calibration. To this end, we propose a novel model, the noise-contrastive variational information bottleneck network (NC-VIBN), which builds upon the VIBN to improve uncertainty estimation. Instead of using distribution matching as the primary source of regularization, our model utilizes it to define a loss term that explicitly encourages high predictive entropy *only* for uninformative

samples from the latent prior. Additionally, we account for weight uncertainties in the decoder and use $L_2$-normalization before computing the latent embeddings to further alleviate the described problems while improving the calibration and generalization capabilities of the model.

**We make the following contributions:** *(i)* We empirically show that models that explicitly regularize the prediction confidence make it harder to distinguish between correct and incorrect predictions based on the estimated uncertainty. *(ii)* We link the VIBN to these methods and find that it suffers from the same behavior due to the implicit $L_2$-regularization through the latent KL-divergence term. *(iii)* We circumvent these ill effects regarding separability of correct and incorrect predictions by proposing a noise-contrastive loss term that utilizes distribution matching in the latent space and, combined with architectural refinements, leads to improved separability, calibration, *and* accuracy. *(iv)* Our results show that our proposed model also leads to improved accuracy and calibration in the presence of distributional shift introduced by image corruptions.

## 2 RELATED WORK

**Information Bottleneck.** The information bottleneck principle has been proposed by Tishby et al. (1999) as means to analyze generalization for deep neural networks (Tishby & Zaslavsky, 2015) and relies on computing the mutual information of input, output, and intermediate variables. Since these quantities are in general intractable, variational approximations of the mutual information terms have been introduced (Alemi et al., 2017; Achille & Soatto, 2018). These variational approximations are also able to overcome problems of the information bottleneck objective for deterministic representations (Amjad & Geiger, 2019) and share some of the benefits of Bayesian models (Alemi et al., 2020). Further, as shown by Alemi et al. (2018), variational information bottleneck networks can improve calibration, which we want to further improve with our method.

**Noise-contrastive estimation.** Noise-contrastive estimation (Gutmann & Hyvärinen, 2010) is an estimation method for parameterized densities, which is based on discriminating between real and artificial data by logistic regression using the log-density functions. Inspired by noise-contrastive estimation, Hafner et al. (2020) propose noise-contrastive priors as data space priors that encourage uncertain predictions at the boundary of the training data for regression tasks by minimizing the KL-divergence between a high-variance Gaussian and the predicted output distribution for perturbed training data points. We focus on classification instead.

**Uncertainty estimation.** Bayesian neural networks (BNNs) are a popular and theoretically well-founded tool for uncertainty estimation. A multitude of methods have been proposed to approximate the intractable weight posterior, including variational inference (Blundell et al., 2015; Graves, 2011), Markov chain Monte Carlo methods (Welling & Teh, 2011), Laplace approximation (MacKay, 1992; Ritter et al., 2018), and assumed density filtering (Hernández-Lobato & Adams, 2015), as well as approximate variational inference methods based on the dropout (Srivastava et al., 2014) regularization scheme (Gal & Ghahramani, 2016). In the last years, these methods have been successfully scaled to larger models (Dusenberry et al., 2020; Heek & Kalchbrenner, 2019; Maddox et al., 2019; Osawa et al., 2019; Zhang et al., 2020). Deep ensembles (Lakshminarayanan et al., 2017) have also been used for uncertainty estimation and can be interpreted as Bayesian model averaging. Since model averaging with respect to the approximate posterior requires multiple forward passes, BNNs incur a substantial computational overhead. To lighten this, there has been an interest in Bayesian last layer approaches (van Amersfoort et al., 2021; Kristiadi et al., 2020; Liu et al., 2020; Riquelme et al., 2018; Snoek et al., 2015; Wilson et al., 2016). Our approach similarly employs a Bayesian treatment only for the last (few) layers, however combining it with the information bottleneck principle and a noise-contrastive loss. An alternative approach for estimating uncertainty in classification networks is parameterizing the more expressive Dirichlet distribution (Gast & Roth, 2018; Joo et al., 2020; Malinin & Gales, 2018; Sensoy et al., 2018) instead of the categorical distribution at the output layer. We instead gain additional expressiveness by modelling the latent space distributions.

**Related methods in out-of-distribution detection.** Lee et al. (2018) proposed to train confidence-calibrated classifiers by using a generative adversarial network (GAN) that learns to generate samples at the data boundary. The generator is trained to generate data points, which are hard to separate from in-distribution data by the discriminator while given an almost uniform labeling by the classifier. In contrast, Sricharan & Srivastava (2018) train the generator to produce low-entropy in-distribution samples while requiring the classifier to maximize the entropy of those samples.

## 3 VARIATIONAL APPROXIMATION OF THE INFORMATION BOTTLENECK

We begin by recapitulating two variational approximations of the information bottleneck, the deep variational information bottleneck (Alemi et al., 2017) and information dropout (Achille & Soatto, 2018), which we will use to explain the behavior of such models and to build our own model upon.

**Deep variational information bottleneck.** The information bottleneck was first introduced (Tishby et al., 1999; Tishby & Zaslavsky, 2015) to find a low-complexity representation $Z$ depending on a feature vector $X$ that maximizes the mutual information with a target variable $Y$. To constrain the complexity of $Z$, the mutual information between $X$ and $Z$ is bounded, resulting in a maximization problem with inequality constraints. Alemi et al. (2017) proposed to use a variational approximation of the mutual information terms of the Lagrangian with Lagrange multiplier $\beta$, resulting in the objective

$$\min_{\phi,\psi} \frac{1}{N} \sum_{n=1}^{N} \mathbb{E}_{p_\phi(z|x_n)} \big[ - \log q_\psi(y_n|z) \big] + \beta D_{\mathrm{KL}} \big[ p_\phi(z|x_n) \| r(z) \big], \tag{1}$$

where the stochastic encoder $p_\phi(z|x)$ and decoder $q_\psi(y|z)$ are modeled by neural networks, parameterized by $\phi$ and $\psi$ respectively, and $r(z)$ is a variational approximation of the marginal distribution $p(z) = \int p_\phi(z|x)p(x)\,\mathrm{d}x$ of $z$. To draw a parallel to the variational inference literature (Kingma & Welling, 2014), $r(z)$ is also referred to as the latent prior. Alemi et al. (2017) assume $r(z)$ to be a standard Gaussian and model the distribution of the latent encodings as Gaussians with diagonal covariance, resulting in

$$D_{\mathrm{KL}} \big[ p_\phi(z|x_n) \| r(z) \big] = \frac{1}{2} \sum_i \Big[ - \log \sigma^2_{z^i|x_n} + \sigma^2_{z^i|x_n} + \mu^2_{z^i|x_n} - 1 \Big], \tag{2}$$

where $\mu_{z^i|x_n}$ and $\sigma^2_{z^i|x_n}$ are the component-wise mean and variance of the latent embedding $p_\phi(z|x_n)$ of $x_n$, estimated by the encoder network. The decoder is a softmax classification network $h_\psi$, predicting class probability vectors from $z$, i.e. $q_\psi(y|z) = \mathrm{Cat}(y|h_\psi(z))$.

**Information dropout (IDO).** Achille & Soatto (2018) proposed an alternative approach, where the encoder network $g_\phi$ predicts a non-negative vector $\mu_x$ as well as $\alpha_x$, parameterizing the log-normal distribution $\log \mathcal{N}(0, \alpha_x^2)$. The distribution of the latent encodings is modeled as $z \sim \mu_x \odot \epsilon$ with $\epsilon \sim \log \mathcal{N}(0, \alpha_x^2)$. They show that if $\mu_x$ is the output of ReLU units and the latent prior $r(z)$ is chosen to be a mixture of the delta distribution at 0 and a log-uniform distribution, i.e. $r(z) \propto q\delta_0(z) + c/z$, the KL-divergence is given as

$$D_{\mathrm{KL}} \big[ p_\phi(z|x) \| r(z) \big] = \begin{cases} - \log q & \mu_x = 0 \\ -\mathrm{H} \big[ p_{\alpha_x}(\log \epsilon) \big] + \log c & \mu_x > 0, \end{cases} \tag{3}$$

where the entropy term $\mathrm{H}[p_{\alpha_x}(\log \epsilon)]$ is given by $\log \alpha_x$ for $\epsilon \sim \log \mathcal{N}(0, \alpha_x^2)$ up to an additive constant. In the original formulation of Achille & Soatto (2018), the mean of $\epsilon$ grows with $\alpha_x$, resulting in a higher level of saturation of the softmax outputs, hence in overconfidence. Note that if $\epsilon$ is log-normal distributed, $\log \epsilon$ is normal distributed and the entropy does not depend on its mean. Therefore, we here instead employ the mean-corrected log-normal distribution $\log \mathcal{N}(-\frac{1}{2}\alpha_x^2, \alpha_x^2)$ so that $\mathbb{E}_{p_{\alpha_x}(\epsilon)}[\mu_x \odot \epsilon] = \mu_x$ without changes to the KL-divergence.

## 4 UNCERTAINTY QUANTIFICATION UNDER OUTPUT DISTRIBUTION REGULARIZATION

A frequently used metric to assess uncertainty estimation is the expected calibration error (Guo et al., 2017) or related calibration metrics, which measure how well the prediction confidence coincides with the prediction accuracy. Methods that achieve better calibration by output distribution regularization include label smoothing (Müller et al., 2019; Szegedy et al., 2016), regularization of the predictive entropy (Pereyra et al., 2017) for the categorical distribution, or evidential deep learning (Sensoy et al., 2018) and belief matching (Joo et al., 2020) for the Dirichlet distribution. Alternatively, the model can be regularized in function or distribution spaces prior to the final softmax layer (Joo et al., 2020), and in the simplest case results in norm penalties for the predicted logits.

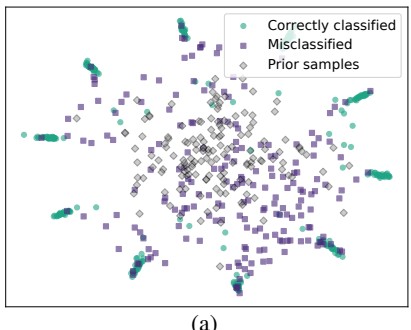
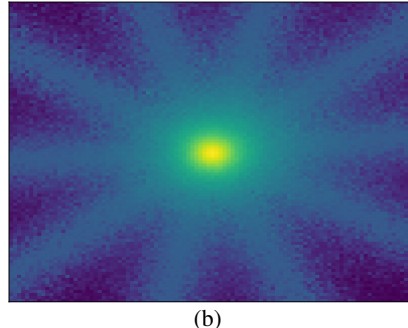

(a)                                                      (b)

Figure 2: *Embeddings and samples from the latent prior (a) and assigned predictive entropy (b) for a NC-VIBN model* with a two-dimensional information bottleneck trained on CIFAR-10. Darker areas correspond to lower entropy. By employing the proposed noise-contrastive loss, embeddings of incorrectly classified examples, which are harder to distinguish from samples of the latent prior, are assigned higher entropy, allowing for a better separability of correct and incorrect predictions based on the predictive entropy.

However, these calibration metrics actually do *not* indicate how well correct predictions can be separated from incorrect ones based on some uncertainty estimate like the predictive entropy. In fact, a model predicting with a confidence equal to the expected accuracy on in-domain data for both correct classifications and misclassifications would be perfectly calibrated according to the expected calibration error, but the confidence of a prediction would give us no information about an example being correctly or incorrectly classified (Ding et al., 2020). A suitable method to examine how easily correct and incorrect predictions can be discriminated, is to analyze how the predictive performance behaves under the rejection of uncertain predictions (Ding et al., 2020; Nadeem et al., 2009). As can be seen in Fig. 1, models that are trained with an objective that regularizes the output distribution perform significantly worse at assigning incorrect predictions a comparatively higher uncertainty and rejecting them early than the deterministic model. We argue that this is because these methods *indiscriminately* regularize the confidence of correct and incorrect predictions. More experimental evidence is presented in Sec. 6.1.

## 5   NC-VIBN – NOISE-CONTRASTIVE VARIATIONAL INFORMATION BOTTLENECK NETWORKS

We can observe in Fig. 1 that, similarly to the regularization methods described in the previous section, standard VIBN models also suffer from worse separability of correct and incorrect predictions. However, we can make use of distribution matching in the latent space to remedy this problem while still benefiting from the improved generalization and calibration of VIBNs. To improve uncertainty estimation with variational bottleneck networks, we here propose a *noise-contrastive loss term* that encourages the decoder network to assign high predictive entropy to samples that are closely aligned to the latent prior. We further propose to use instance-level $L_2$-regularization before computing the parameters of the latent distribution to reduce the sensitivity to the magnitude of input variables and also learn the uncertainty of the decoder parameters using variational inference.

### 5.1   LEARNING WEIGHT UNCERTAINTIES FOR THE CLASSIFIER

To improve the ability of VIBNs to quantify uncertainty, we first propose to learn a weight distribution $q(\psi|\theta)$ for the decoder via mean field variational inference. A similar approach for variational autoencoders has been explored by Daxberger & Hernández-Lobato (2019), who learn an ensemble of generative models by approximating the decoder posterior with Markov chain Monte Carlo methods. We assume that we have a dataset $\mathcal{D} = (z_n, y_n)$ such that

$$- \log p(y_n|z_n, \psi) = \mathbb{E}_{p_\phi(z|x_n)}[- \log q_\psi(y_n|z)] + \text{const} \qquad (4)$$

and a weight prior $p(\psi) = \mathcal{N}(\psi|0, \tau^{-1}I)$. Now, minimizing the KL-divergence between the mean field Gaussian variational approximation and the true posterior is equivalent to solving

$$\min_{\theta} \frac{1}{N} \sum_{n=1}^{N} \mathbb{E}_{q(\psi|\theta)} \Big[ \mathbb{E}_{p_\phi(z|x_n)} \big[ -\log q_\psi(y_n|z) \big] \Big] + \frac{1}{N} D_{\mathrm{KL}} \big[ q(\psi|\theta) \| p(\psi) \big]. \tag{5}$$

This loss function will substitute the first term in the VIBN objective in equation 1.

### 5.2 THE NOISE-CONTRASTIVE LOSS TERM

Next, we will utilize the distributional alignment of the variational information bottleneck to formulate a noise-constrastive loss in the latent space. This has multiple benefits as opposed to formulating the noise-contrastive loss in the input space: We do not have to approximate the complex distribution of images in the high-dimensional input space and only have to evaluate the smaller decoder network to obtain predictions for samples from this distribution.

To motivate how we can improve the separability of correct and incorrect examples by the predictive entropy, consider Fig. 2a. We can observe distinct clusters for the latent embeddings of correctly classified examples, while incorrectly classified examples for the most part are embedded in-between these clusters, resembling samples from the latent prior. Moreover, we note that embeddings of hard examples tend to be aligned with the latent prior, since the KL-term of the VIBN objective (equation 1), which regularizes the distance of the latent embedding from the prior, is class-agnostic and thus easier to minimize than the negative log-likelihood term for such hard examples. To now prevent overconfident predictions of the decoder network in these regions of the latent space, we propose a noise-contrastive loss that maximizes the predictive entropy for samples generated from the latent prior, resulting in the objective

$$\max_{\theta} \mathbb{E}_{q(\psi|\theta)} \Big[ \mathbb{E}_{r(z)} \big[ \mathrm{H}[q_\psi(y|z)] \big] \Big], \tag{6}$$

where the predictive entropy of the categorical distribution estimated by the classifier is given by $H(q_\psi(y|z)) = -\sum_{c=1}^{C} q_\psi(y = c|z) \log q_\psi(y = c|z)$.

Note that embeddings for easier examples, in contrast, are mapped into regions of the latent space away from the high-density regions of the latent prior in order to enable the decoder to separate them from prior samples, yet prior samples are assigned low confidence due to the entropy maximizing noise-contrastive loss. This separation helps the decoder to assign high confidence to the embeddings of easier examples, so that the expected negative log-likelihood term of the VIBN objective can be minimized. The effect of the interplay of the VIBN objective and the noise-contrastive loss can be seen in Fig. 2b, where high entropy is assigned to the regions of the latent space that are populated with embeddings of incorrectly classified examples, while regions where the embeddings of correctly assigned examples are located are assigned lower entropy (see appendix B.1 for a comparison with the latent space of a standard VIBN). As a result, we can preserve the separability of correct and incorrect examples by entropy-based uncertainty estimates while still benefiting from the improved calibration and regularization of VIBN networks.

Putting everything together and approximating expectations via sampling, we obtain the following loss function:

$$L(\phi, \theta) = \frac{1}{N} \sum_{n=1}^{N} \left( \frac{1}{S} \sum_{s=1}^{S} -\log q_{\psi^s}(y_n|z_{x_n}^s) + \beta D_{\mathrm{KL}} \big[ p_\phi(z|x_n) \| r(z) \big] \right) \tag{7}$$
$$+ \frac{1}{N} D_{\mathrm{KL}} \big[ q(\psi|\theta) \| p(\psi) \big] - \frac{1}{S} \sum_{s=1}^{S} \mathrm{H} \big[ q_{\psi^s}(y|z^s) \big],$$

where $S$ samples $z_{x_n}^s$ are drawn from $p(z|x_n)$ and $\psi^s$ are drawn from $q(\psi|\theta)$. The first loss term maximizes an empirical approximation of the expected log-likelihood, while the second loss term aligns the distribution of the latent embeddings with the latent prior. Further, the third loss term is a result of the variational approximation of the classifier's weight posterior distribution and penalizes divergence from the weight prior. Finally, the last loss term is our noise-contrastive loss, encouraging high entropy for noisy latent prior samples.

## 5.3 TEST-TIME NORMALIZATION

Hein et al. (2019) describe how the linear extrapolation behavior of ReLU networks can lead to overconfidence for data points that differ from the training data in terms of magnitude or, in case of image data, in pixel intensity. Similarly, for our encoder network this behavior can result in embeddings into regions that are far away from the latent prior and assigned high certainty by the decoder network for these data points. To prevent this and decouple the predicted distribution parameters from the intensity of pixel values, we add a test-time normalization step to the intermediate variables before computing the latent embeddings. Specifically, we apply $L_2$-normalization to the input $\tilde{x}$ of the last residual block of the encoder network, mapping

$$\tilde{x} \mapsto \tilde{x}/\sqrt{\textstyle\sum_{j=1}^{M} \tilde{x}_j^2 + \epsilon}, \tag{8}$$

where $\epsilon > 0$ is a small constant added for numerical stability and $M$ is the dimension of $\tilde{x}$.

## 5.4 IMPLEMENTATION DETAILS

**Encoder and classifier architecture.** We adjust the ResNet architecture (He et al., 2016) for our model by using the last residual block, consisting of two convolutional layers for smaller ResNets and three convolutional layers for larger ResNets, and the fully connected layer as the classifier (decoder) network; the remaining layers form the encoder network. The channels of the last layer of the encoder are split to predict $\mu_x$ and $\sigma_x^2$ or $\alpha_x$ for the VIBN or IDO models, respectively. The width of the classifier network is accordingly reduced by half, resulting in a lower total number of parameters compared to the respective (deterministic) ResNet baseline, even though additional parameters representing the weight variances of the decoder are required.

**Parameterizing the latent distribution.** For the VIBN, we predict $\log \sigma_x^2$ and map it to $\sigma_x^2$ using the exponential function. For IDO, we follow Achille & Soatto (2018) and constrain $\alpha_x$ to be smaller than $\alpha_{\max}$. For simplicity, we chose $\alpha_{\max} = 1$. Instead of predicting $\alpha_x$ directly, the unrestricted variable $\hat{\alpha}_x$ is predicted and mapped to $\alpha_x = \alpha_{\max} \cdot \sigma(\hat{\alpha}_x)$, where $\sigma(\cdot)$ is the sigmoid function.

**Sampling from the latent distribution and prior.** To sample from the latent distribution in case of the VIBN, we use $z_x = \mu_x + \sigma_x \odot \epsilon$, where $\epsilon \sim \mathcal{N}(0, I)$. Samples from the latent prior can be directly generated from $\mathcal{N}(0, I)$. For IDO, we again generate samples $\epsilon$ from $\mathcal{N}(0, I)$ and map them to $\mu_x \odot \exp(\alpha_x \odot \epsilon - \alpha_x^2/2)$ to sample from the corresponding latent distribution. We generate samples from a truncation of the latent prior by sampling $u$ from a uniform distribution on $[-2, 2]$ and $b$ from a Bernoulli distribution with parameter $q = 0.1$ and computing $z = b \odot \exp(u)$.

**Weight uncertainties for the classifier.** We initialize the weight variances to $\rho$ times the prior variance; more details can be found in appendix A. We use the local reparameterization trick (Kingma et al., 2015) to sample from the classifier weight distributions.

**Rescaling the loss terms.** We do not introduce any additional hyperparameters for rescaling the loss terms except for the latent KL-divergence inherited from VIBN. Further, we apply a linear warm-up schedule to all loss terms except the expected negative log-likelihood to emulate training of a deterministic network in the early stages.

# 6 EXPERIMENTS

We begin by comparing the proposed noise-contrastive variational information bottleneck networks with fixed standard Gaussian prior (NC-VIBN) and learnable Gaussian mixture prior (NC-VIBN-MP) as well as information dropout networks (NC-IDO) to their counterparts VIBN (Alemi et al., 2017), VIBN-MP, and IDO (Achille & Soatto, 2018), as well as to a deterministic network and a BNN using MC Dropout (Gal & Ghahramani, 2016) as baselines. For fair comparison in terms of inference time, we apply dropout only to layers that need to be resampled multiple times for the information bottleneck networks. For the mixture prior experiments, we consider a mixture of 32 Gaussian components with diagonal covariance matrices and allow learning the means, variances, and mixture weights for every component. During evaluation, we use the same number of samples ($S = 16$) for MC Dropout and all information bottleneck networks. We train models based on ResNet18 (He et al., 2016) on CIFAR-10, CIFAR-100 (Krizhevsky & Hinton, 2009), and Tiny ImageNet, a downsampled version of ImageNet (Deng et al., 2009) containing 200 classes. The results can be found in Tab. 1.

Table 1: *Evaluation on CIFAR-10, CIFAR-100, and Tiny ImageNet.* We report the misclassification rate (MCR), negative log-likelihood (NLLH), expected calibration error (ECE), the area under the misclassification-rejection curve (AUMRC), the misclassification rate (MR) at different rejection levels as well as the mean confidence (MC) for correct and incorrect predictions. All values but NLLH are given as percentages.

| | | MCR | NLLH | ECE | AUMRC | MR5% | MR10% | MR25% | MC Corr. | MC Incorr. |
|---|---|---|---|---|---|---|---|---|---|---|
| CIFAR-10 | Deterministic | $5.37_{\pm0.17}$ | $0.237_{\pm0.012}$ | $3.22_{\pm0.19}$ | $\mathbf{0.53_{\pm0.02}}$ | $3.11_{\pm0.16}$ | $1.83_{\pm0.11}$ | $\mathbf{0.33_{\pm0.07}}$ | $98.57_{\pm0.34}$ | $82.11_{\pm0.55}$ |
| | MC Dropout | $5.67_{\pm0.22}$ | $0.243_{\pm0.006}$ | $3.52_{\pm0.13}$ | $0.56_{\pm0.01}$ | $3.37_{\pm0.06}$ | $1.94_{\pm0.03}$ | $0.47_{\pm0.06}$ | $98.75_{\pm0.04}$ | $82.11_{\pm0.89}$ |
| | IDO | $5.67_{\pm0.08}$ | $0.238_{\pm0.004}$ | $3.30_{\pm0.08}$ | $0.59_{\pm0.04}$ | $3.42_{\pm0.05}$ | $1.99_{\pm0.05}$ | $0.41_{\pm0.07}$ | $98.41_{\pm0.23}$ | $81.15_{\pm0.67}$ |
| | VIBN | $5.11_{\pm0.12}$ | $0.206_{\pm0.004}$ | $2.03_{\pm0.16}$ | $0.92_{\pm0.06}$ | $2.94_{\pm0.17}$ | $1.76_{\pm0.08}$ | $0.89_{\pm0.03}$ | $97.97_{\pm0.05}$ | $76.48_{\pm1.47}$ |
| | VIBN-MP | $5.13_{\pm0.24}$ | $0.180_{\pm0.006}$ | $\mathbf{1.08_{\pm0.09}}$ | $0.88_{\pm0.18}$ | $3.09_{\pm0.23}$ | $1.78_{\pm0.11}$ | $0.73_{\pm0.11}$ | $97.03_{\pm0.15}$ | $73.45_{\pm0.87}$ |
| | NC-IDO *(ours)* | $5.09_{\pm0.19}$ | $0.174_{\pm0.006}$ | $1.90_{\pm0.14}$ | $0.53_{\pm0.06}$ | $2.92_{\pm0.17}$ | $1.58_{\pm0.12}$ | $0.41_{\pm0.04}$ | $97.93_{\pm0.05}$ | $75.11_{\pm0.95}$ |
| | NC-VIBN *(ours)* | $4.93_{\pm0.12}$ | $0.165_{\pm0.003}$ | $1.28_{\pm0.21}$ | $0.57_{\pm0.06}$ | $2.83_{\pm0.15}$ | $1.52_{\pm0.10}$ | $0.45_{\pm0.06}$ | $97.48_{\pm0.11}$ | $71.99_{\pm1.08}$ |
| | NC-VIBN-MP *(ours)* | $\mathbf{4.74_{\pm0.10}}$ | $\mathbf{0.164_{\pm0.003}}$ | $1.16_{\pm0.15}$ | $0.56_{\pm0.04}$ | $\mathbf{2.76_{\pm0.06}}$ | $\mathbf{1.49_{\pm0.09}}$ | $0.45_{\pm0.04}$ | $97.55_{\pm0.12}$ | $72.00_{\pm0.75}$ |
| CIFAR-100 | Deterministic | $26.22_{\pm1.15}$ | $1.146_{\pm0.0123}$ | $10.92_{\pm0.23}$ | $7.73_{\pm0.14}$ | $23.32_{\pm0.28}$ | $20.73_{\pm0.28}$ | $13.40_{\pm0.25}$ | $92.28_{\pm0.10}$ | $63.39_{\pm0.42}$ |
| | MC Dropout | $25.75_{\pm0.12}$ | $1.358_{\pm0.0283}$ | $15.35_{\pm0.30}$ | $\mathbf{7.35_{\pm0.11}}$ | $22.83_{\pm0.20}$ | $20.17_{\pm0.20}$ | $\mathbf{12.58_{\pm0.26}}$ | $95.36_{\pm0.08}$ | $72.97_{\pm0.84}$ |
| | IDO | $26.10_{\pm0.29}$ | $1.387_{\pm0.016}$ | $13.55_{\pm0.16}$ | $7.63_{\pm0.05}$ | $23.18_{\pm0.34}$ | $20.43_{\pm0.33}$ | $12.93_{\pm0.35}$ | $93.94_{\pm0.12}$ | $69.26_{\pm0.25}$ |
| | VIBN | $26.19_{\pm0.49}$ | $1.310_{\pm0.033}$ | $8.50_{\pm0.78}$ | $8.29_{\pm0.26}$ | $23.62_{\pm0.46}$ | $21.17_{\pm0.67}$ | $14.10_{\pm0.58}$ | $90.24_{\pm0.28}$ | $59.50_{\pm1.32}$ |
| | VIBN-MP | $26.37_{\pm0.24}$ | $1.118_{\pm0.011}$ | $\mathbf{4.93_{\pm0.45}}$ | $8.91_{\pm0.05}$ | $23.93_{\pm0.20}$ | $21.59_{\pm0.13}$ | $14.99_{\pm0.25}$ | $83.87_{\pm1.13}$ | $45.57_{\pm1.94}$ |
| | NC-IDO *(ours)* | $25.35_{\pm0.22}$ | $1.117_{\pm0.035}$ | $9.06_{\pm0.43}$ | $7.38_{\pm0.10}$ | $22.47_{\pm0.31}$ | $\mathbf{20.01_{\pm0.10}}$ | $12.84_{\pm0.25}$ | $91.30_{\pm0.23}$ | $61.28_{\pm0.63}$ |
| | NC-VIBN *(ours)* | $25.16_{\pm0.25}$ | $\mathbf{1.009_{\pm0.022}}$ | $5.58_{\pm0.26}$ | $7.97_{\pm0.18}$ | $22.58_{\pm0.26}$ | $20.27_{\pm0.32}$ | $13.98_{\pm0.22}$ | $87.30_{\pm0.60}$ | $51.33_{\pm2.04}$ |
| | NC-VIBN-MP *(ours)* | $\mathbf{25.07_{\pm0.38}}$ | $1.044_{\pm0.01}$ | $6.72_{\pm0.38}$ | $7.83_{\pm0.14}$ | $\mathbf{22.45_{\pm0.36}}$ | $20.56_{\pm1.05}$ | $13.67_{\pm0.45}$ | $88.00_{\pm0.33}$ | $51.19_{\pm0.70}$ |
| Tiny ImageNet | Deterministic | $43.09_{\pm0.25}$ | $2.184_{\pm0.118}$ | $18.26_{\pm2.23}$ | $17.93_{\pm0.24}$ | $40.64_{\pm0.27}$ | $38.20_{\pm0.32}$ | $30.63_{\pm0.17}$ | $87.64_{\pm1.52}$ | $58.70_{\pm3.42}$ |
| | MC Dropout | $\mathbf{41.13_{\pm0.48}}$ | $1.838_{\pm0.029}$ | $10.05_{\pm0.49}$ | $\mathbf{16.44_{\pm0.27}}$ | $\mathbf{38.66_{\pm0.66}}$ | $\mathbf{35.94_{\pm0.23}}$ | $\mathbf{28.35_{\pm0.42}}$ | $86.04_{\pm1.39}$ | $77.51_{\pm0.76}$ |
| | IDO | $43.40_{\pm0.21}$ | $2.081_{\pm0.10}$ | $16.14_{\pm2.47}$ | $18.32_{\pm0.18}$ | $41.02_{\pm0.21}$ | $38.76_{\pm0.23}$ | $31.10_{\pm0.46}$ | $85.73_{\pm1.84}$ | $55.78_{\pm3.49}$ |
| | VIBN | $43.23_{\pm0.28}$ | $2.092_{\pm0.008}$ | $11.99_{\pm0.18}$ | $18.49_{\pm0.22}$ | $40.84_{\pm0.30}$ | $38.46_{\pm0.23}$ | $31.27_{\pm0.26}$ | $82.26_{\pm0.14}$ | $51.03_{\pm0.21}$ |
| | VIBN-MP | $42.83_{\pm0.39}$ | $1.843_{\pm0.008}$ | $6.67_{\pm0.44}$ | $18.11_{\pm0.23}$ | $40.47_{\pm0.39}$ | $38.10_{\pm0.45}$ | $30.78_{\pm0.32}$ | $78.50_{\pm0.35}$ | $44.23_{\pm0.71}$ |
| | NC-IDO *(ours)* | $42.09_{\pm0.51}$ | $1.784_{\pm0.033}$ | $\mathbf{4.90_{\pm0.64}}$ | $17.62_{\pm0.51}$ | $39.65_{\pm0.53}$ | $37.01_{\pm0.57}$ | $30.01_{\pm0.58}$ | $77.78_{\pm1.02}$ | $42.10_{\pm0.76}$ |
| | NC-VIBN *(ours)* | $41.82_{\pm0.12}$ | $\mathbf{1.756_{\pm0.040}}$ | $6.60_{\pm2.06}$ | $17.57_{\pm0.21}$ | $39.45_{\pm0.14}$ | $37.15_{\pm0.17}$ | $29.85_{\pm0.23}$ | $79.61_{\pm1.61}$ | $44.01_{\pm2.82}$ |
| | NC-VIBN-MP *(ours)* | $41.46_{\pm0.20}$ | $1.781_{\pm0.008}$ | $9.35_{\pm0.52}$ | $16.91_{\pm0.16}$ | $39.08_{\pm0.23}$ | $36.76_{\pm0.33}$ | $29.26_{\pm0.27}$ | $82.39_{\pm0.58}$ | $47.36_{\pm0.81}$ |

Table 2: *Comparison to output regularization methods and DUE (van Amersfoort et al., 2021) on CIFAR-10.* Regularization methods are more accurate and better calibrated than the deterministic model, but perform worse at separating correct and incorrect predictions via their predictive entropy, resulting in a higher AUMRC. Our NC-VIBN clearly outperforms these previous methods.

| | MCR | NLLH | ECE | AUMRC | MR5% | MR10% | MR25% | MC Corr. | MC Incorr. |
|---|---|---|---|---|---|---|---|---|---|
| Deterministic | $5.37_{\pm0.17}$ | $0.237_{\pm0.012}$ | $3.22_{\pm0.19}$ | $\mathbf{0.53_{\pm0.02}}$ | $3.11_{\pm0.16}$ | $1.83_{\pm0.11}$ | $\mathbf{0.33_{\pm0.07}}$ | $98.57_{\pm0.34}$ | $82.11_{\pm0.55}$ |
| Entropy Regularization | $5.20_{\pm0.17}$ | $0.201_{\pm0.004}$ | $2.01_{\pm0.24}$ | $1.49_{\pm0.12}$ | $3.11_{\pm0.19}$ | $1.97_{\pm0.18}$ | $1.39_{\pm0.20}$ | $94.84_{\pm0.19}$ | $73.13_{\pm0.90}$ |
| Logit $L_2$-Regularization | $5.33_{\pm0.30}$ | $0.208_{\pm0.006}$ | $1.74_{\pm0.21}$ | $1.35_{\pm0.03}$ | $3.11_{\pm0.22}$ | $1.85_{\pm0.04}$ | $0.98_{\pm0.05}$ | $97.40_{\pm0.04}$ | $78.45_{\pm0.84}$ |
| Label Smoothing | $5.15_{\pm0.14}$ | $0.209_{\pm0.005}$ | $\mathbf{1.28_{\pm0.18}}$ | $1.57_{\pm0.18}$ | $3.16_{\pm0.17}$ | $1.97_{\pm0.09}$ | $1.14_{\pm0.13}$ | $96.73_{\pm0.13}$ | $78.10_{\pm0.16}$ |
| Dirichlet Belief Matching | $5.15_{\pm0.11}$ | $0.197_{\pm0.004}$ | $1.47_{\pm0.15}$ | $1.07_{\pm0.19}$ | $3.03_{\pm0.17}$ | $1.76_{\pm0.04}$ | $0.88_{\pm0.11}$ | $96.73_{\pm0.37}$ | $77.71_{\pm0.85}$ |
| DUE | $5.27_{\pm0.28}$ | $0.195_{\pm0.007}$ | $2.24_{\pm0.21}$ | $0.76_{\pm0.03}$ | $3.06_{\pm0.19}$ | $1.83_{\pm0.11}$ | $0.68_{\pm0.07}$ | $97.92_{\pm0.16}$ | $78.79_{\pm0.77}$ |
| NC-VIBN *(ours)* | $\mathbf{4.93_{\pm0.12}}$ | $\mathbf{0.165_{\pm0.003}}$ | $1.28_{\pm0.21}$ | $0.57_{\pm0.06}$ | $\mathbf{2.83_{\pm0.15}}$ | $\mathbf{1.52_{\pm0.10}}$ | $0.45_{\pm0.06}$ | $97.48_{\pm0.11}$ | $71.99_{\pm1.08}$ |

Next, we compare against several baseline approaches for regularizing the output distribution of DNNs in Tab. 2. We report the average performance and standard deviation of four training runs.

We use the misclassification rate (MCR) to assess how well a method works as a regularizer and improves generalization. To evaluate the uncertainty quantification, we report the negative log-likelihood (NLLH), the expected calibration error (ECE; Guo et al., 2017), the area under the misclassification-rejection curve (AUMRC; Nadeem et al., 2009), and the misclassfication rate at certain rejection rates. Training details, a detailed description of the evaluation metrics (Sec. A), as well as further experiments (Sec. B) can be found in the appendix.

## 6.1 IN-DISTRIBUTION CLASSIFICATION ACCURACY AND UNCERTAINTY QUANTIFICATION

**Regularization of the output distribution and intermediate variables.** We first look in detail at methods that regularize the categorical distribution, such as entropy regularization (Pereyra et al., 2017) and label smoothing (Müller et al., 2019; Szegedy et al., 2016), or the Dirichlet output distribution, in particular Dirichlet belief matching (Joo et al., 2020), as well as methods that regularize the intermediate variables, such as logit $L_2$-regularization (Joo & Chung, 2020). Note that the KL-divergence term of the VIBN also implicitly regularizes the $L_2$-norm of the latent variables. From Tab. 2 we observe higher accuracy compared to the deterministic baseline, which shows that these baselines indeed work well as regularization and improve generalization. By controlling the

regularization of the output distribution or intermediate variables, calibration can be improved and the confidence for incorrect predictions is reduced. However, the confidence of correct predictions is decreased as well. Hence differentiating between correct and incorrect predictions based on the predictive entropy becomes harder, as indicated by the MR-curve (Fig. 1). While this effect is more pronounced at higher rejection rates, it still significantly impacts the area under the MR-curve, increasing it from 0.005 for the deterministic baseline on CIFAR-10 to above 0.010 for these models. The benefits of our NC-VIBN become apparent when comparing the mean confidence on correct and incorrect predictions to output regularization methods (Tab. 2). Even though these methods using explicit regularization can reduce the confidence of incorrect predictions as desired, our NC-VIBN still shows lower confidence on incorrect predictions while not impacting the confidence of correct predictions as much. As a consequence, it is better calibrated than output regularization methods without the increase in AUMRC, while being more accurate at the same time.

Next, we compare our results with DUE (van Amersfoort et al., 2021). There has been a recent interest in uncertainty estimation methods that use Gaussian processes and RBF kernels on the network output, for example SNGP (Liu et al., 2020), DUQ (van Amersfoort et al., 2020), and the aforementioned and most recent DUE. These methods enforce Lipschitz constraints on the neural network to ensure that the feature distance in the output space contains information about feature distance in the input space, avoid feature collapse, and improve the ability to detect out-of-distribution data points. For in-distribution data, as can be seen in Tab. 2, DUE offers benefits regarding the misclassification rate, NLLH, and ECE, although not as significant as our NC-VIBN models. Moreover, the ability of DUE to detect misclassifications based on the uncertainty assigned using information about the output space distance as measured by the AUMRC appears to be considerably worse when compared to the deterministic baseline. Our proposed NC-VIBN outperforms the recent DUE across all metrics.

**Comparison of different latent priors.** We compare three different variational approximations of the information bottleneck objective, IDO, VIBN, and VIBN-MP, in Tab. 1 to understand which of the observed effects are specific to the variational distributions. We find that IDO provides only minor benefits to classification accuracy and no significant benefits to uncertainty quantification as indicated by NLLH and ECE. Yet, we do not see the same rise in AUMRC for IDO as we see for VIBN or VIBN-MP. Since for the variational approximation by IDO these effects are not present, we can conclude that the information bottleneck does not inherently benefit model calibration or harm separability of correct and incorrect predictions. This suggests that the observed behaviour of the VIBN and VIBN-MP arises from the chosen variational distribution and the implicit $L_2$-regularization in the latent KL-divergence term, see equation 2. Note, that the adaptable $L_2$-regularization present for VIBN-MP, while consistently improving calibration, does not avoid the separability problem, only slightly reducing the AUMRC on CIFAR-10 and Tiny ImageNet while increasing it on CIFAR-100.

**Benefits of the NC-VIBN.** As can be seen in Tab. 1, our NC-VIBN improves generalization but also offers a whole number of benefits regarding the uncertainty evaluation metrics, including a significantly better description of the data, as can be seen by lower negative log-likelihood. On CIFAR-10, our proposed model reduces the NLLH of 0.24 for the baseline and 0.21 for VIBN to 0.17. Similar improvements can be obtained on Tiny ImageNet, where the NLLH is decreased from 2.19 and 2.09, respectively, to 1.76. We further see significant improvements regarding calibration (cf. also Fig. 3), lowering the ECE on CIFAR-10 from 0.032 for the deterministic baseline and 0.020 for VIBN to only 0.013, while on Tiny ImageNet the ECE is decreased from 0.183 and 0.120, respectively, to 0.066. Only VIBN-MP with a more complex learnable mixture prior offers slightly better

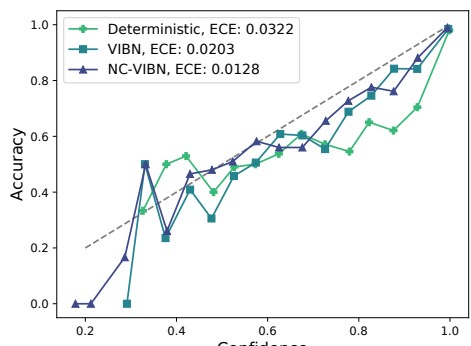

Figure 3: *Calibration curves for the deterministic, VIBN, and NC-VIBN networks on CIFAR-10.* For our NC-VIBN, confidence and accuracy are better aligned.

calibration on CIFAR-10 and -100, while our NC-IDO achieves the best calibration on Tiny ImageNet. Further, we almost recover the AUMRC of the deterministic network on CIFAR-10 and -100, with the misclassification rate only lagging behind at high rejection levels (Fig. 1) while improving it on Tiny ImageNet, achieving the 2nd lowest AUMRC. The same modifications can also be applied to IDO to obtain significant improvements across all metrics.

Table 4: *Image corruptions on CIFAR-10 (top) and CIFAR-100 (bottom) with three different levels of severity.* The NC-VIBN model is consistently more robust to corruptions and offers better uncertainty estimates compared to the deterministic and VIBN baselines.

| | Corruption Level 1 | | | Corruption Level 3 | | | Corruption Level 5 | | |
|---|---|---|---|---|---|---|---|---|---|
| | MCR | NLLH | ECE | MCR | NLLH | ECE | MCR | NLLH | ECE |
| Deterministic | $17.03_{\pm0.39}$ | $0.941_{\pm0.063}$ | $11.76_{\pm0.49}$ | $31.90_{\pm0.55}$ | $2.232_{\pm0.143}$ | $23.12_{\pm0.70}$ | $49.98_{\pm0.74}$ | $4.049_{\pm0.249}$ | $38.05_{\pm1.21}$ |
| VIBN | $16.02_{\pm0.54}$ | $0.790_{\pm0.035}$ | $8.84_{\pm0.38}$ | $30.32_{\pm0.77}$ | $1.791_{\pm0.072}$ | $18.73_{\pm0.68}$ | $48.62_{\pm0.70}$ | $3.193_{\pm0.118}$ | $32.18_{\pm0.68}$ |
| NC-VIBN | $\mathbf{15.03}_{\pm0.46}$ | $\mathbf{0.545}_{\pm0.027}$ | $\mathbf{6.44}_{\pm0.56}$ | $\mathbf{29.31}_{\pm0.75}$ | $\mathbf{1.219}_{\pm0.084}$ | $\mathbf{15.54}_{\pm1.16}$ | $\mathbf{48.16}_{\pm0.74}$ | $\mathbf{2.266}_{\pm0.091}$ | $\mathbf{29.45}_{\pm1.71}$ |
| Deterministic | $40.65_{\pm0.26}$ | $2.047_{\pm0.014}$ | $18.33_{\pm0.10}$ | $55.74_{\pm0.39}$ | $3.336_{\pm0.038}$ | $27.76_{\pm0.25}$ | $71.31_{\pm0.42}$ | $4.955_{\pm0.096}$ | $38.73_{\pm0.70}$ |
| VIBN | $41.64_{\pm0.17}$ | $2.571_{\pm0.074}$ | $16.39_{\pm0.89}$ | $56.52_{\pm0.32}$ | $4.103_{\pm0.154}$ | $25.14_{\pm0.82}$ | $72.07_{\pm0.31}$ | $6.025_{\pm0.244}$ | $35.77_{\pm0.81}$ |
| NC-VIBN | $\mathbf{39.37}_{\pm0.17}$ | $\mathbf{1.768}_{\pm0.025}$ | $\mathbf{10.24}_{\pm0.69}$ | $\mathbf{54.39}_{\pm0.10}$ | $\mathbf{2.764}_{\pm0.0971}$ | $\mathbf{17.51}_{\pm1.77}$ | $\mathbf{70.23}_{\pm0.18}$ | $\mathbf{4.000}_{\pm0.269}$ | $\mathbf{27.02}_{\pm2.91}$ |

The ablation study in Tab. 3 shows that both the deterministic and VIBN baseline models benefit from last-layer variational inference and $L_2$-normalization with respect to classification accuracy and calibration but that they do not substantially improve the separability of correct and incorrect examples as measured by the AUMRC. The noise-contrastive entropy maximization term on its own leads to a far more pronounced reduction in AUMRC while also benefiting the calibration at the cost of a minor drop in accuracy. However, when combined with the other two components the drop in accuracy can be avoided and further significant improvements regarding all metrics can be gained.

Table 3: *Ablations for the NC-VIBN on CIFAR-10.* We compare the effect of $L_2$-normalization and learning weight uncertainties for the last layers via variational inference with the noise-contrastive loss and the full NC-VIBN setup.

| | MCR | NLLH | ECE | AUMRC |
|---|---|---|---|---|
| Deterministic | 5.37 | 0.237 | 3.22 | **0.53** |
| Det. + LL VI & $L_2$-norm. | 5.03 | 0.170 | 1.61 | 0.56 |
| VIBN | 5.11 | 0.206 | 2.03 | 0.92 |
| VIBN + Noise-contrastive loss | 5.23 | 0.187 | 1.41 | 0.74 |
| VIBN + Dec. VI & $L_2$-norm. | 4.95 | 0.179 | 1.55 | 0.87 |
| NC-VIBN | **4.93** | **0.165** | **1.28** | 0.57 |

## 6.2 Estimation Quality Under Distribution Shift Through Image Corruptions

To evaluate robustness and uncertainty quantification in the presence of distributional shift, we use the CIFAR-10-C and CIFAR-100-C datasets (Hendrycks & Dietterich, 2019), which feature 15 different types of common image corruptions at five levels of severity. Since distributional shift will likely cause the latent embeddings to be less aligned with those computed on the clean training data, they may be less distinguishable from samples from the latent prior. In this case, the noise-contrastive entropy loss helps assigning higher entropy to examples misclassified due to distributional shift. The results in Tab. 4 confirm that our proposed NC-VIBN performs better under distributional shift than deterministic and VIBN baselines. It is more robust to these image corruptions, with lower misclassification rates across all levels of corruption severity. Even stronger benefits can be observed for the metrics evaluating uncertainty quantification. The lower NLLH at 2.27 for NC-VIBN compared to 4.95 for the deterministic and 3.19 for the VIBN baseline on the most severe corruptions on CIFAR-10, with similar reductions for other corruption levels for both CIFAR-10 and CIFAR-100, show that our NC-VIBN offers a better explanation for hold-out image data even when corruptions are present. Further, the improved model calibration is retained as the corruption severity increases, e.g. resulting in a significant decrease of the calibration error from above 0.350 for the most severe corruptions on CIFAR-100 to 0.270.

## 7 Conclusion

We propose to improve uncertainty estimation in classification models by building upon the distributional alignment of deep variational information bottleneck networks. Our experiments identify a problem of prior work, being significantly worse at separating correct and incorrect predictions based on the prediction uncertainty, a behavior that can be found in VIBNs as well as other models that regularize the distribution of output or intermediate variables. We address this specifically using a noise-contrastive entropy maximization term, as well as $L_2$-normalization and weight uncertainty in the decoder. Experiments show that our NC-VIBN model can improve generalization and uncertainty estimation at the same time, both on in-domain data and under domain shift by image corruptions.

ETHICS STATEMENT

While our approach can empirically improve the reliability of deep learning systems, failure cases may still arise. However, this is consistent with other black-box uncertainty estimation methods in deep learning. We, therefore, believe that it is important to inform possible users of our as well as other models predicting uncertainty that uncertainty estimates cannot be blindly trusted so that their availability does not induce a false sense of security.

REPRODUCIBILITY STATEMENT

We include training and evaluation code for the noise-contrastive information bottleneck models and baseline models in the supplementary material. Details on hyperparameter selection can be found in appendix A.

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

## A TRAINING AND EVALUATION DETAILS

### A.1 EVALUATION METRICS

For our experiments, we compare the uncertainty estimates of different models with regard to the following metrics:

**Negative log-likelihood.** A low negative log-likelihood (NLLH) indicates that the model of $p(y|x)$ assigns high probability (density) to the correct labels. The metric encourages high confidence for correct classifications but also penalizes assigning high confidence to the incorrect predictions.

**Expected calibration error.** The expected calibration error (ECE) (Guo et al., 2017) is a measure for model calibration, i.e. how well the estimated confidence agrees with the accuracy of predictions of a similar confidence level. We divide the interval $[0, 1]$ into $M = 20$ bins and assign the $N$ predictions into these bins based on their confidence value. Within each bin $B_j$, we compute the prediction accuracy $\text{acc}(j)$ and the average confidence $\text{conf}(j)$. The expected calibration error is then given by $\sum_{j=1}^{M} \frac{|B_j|}{N} |\text{acc}(j) - \text{conf}(j)|$.

**Area under the misclassification-rejection curve.** To evaluate how well a model is able to reduce the misclassification rate by rejecting uncertain predictions, we compute the area under the misclassification-rejection curve (AUMRC) (Nadeem et al., 2009), which is the step function obtained by sequentially excluding the example with the highest predictive entropy and adding a point to the curve where the $x$-coordinate is the fraction of rejected examples and the $y$-coordinate is the misclassification rate among the remaining examples. We also report the misclassification rate at certain rejection thresholds. We chose this metric for evaluating if incorrect predictions are assigned higher uncertainty than correct predictions instead of using the AUROC or AUPR for misclassification detection, because latter metrics are not suited for comparing models with different levels of accuracy (Ding et al., 2020).

### A.2 DATASETS

For CIFAR-10 and CIFAR-100 (Krizhevsky & Hinton, 2009), we split 5 000 images from the training set, which consists of 50 000 images, to use for validation; we report results on the 10 000 test images. For ImageNet (cf. Sec. B.4), we use the ILSVRC2012 (Russakovsky et al., 2015) training split, containing 1.3 million images, to train the networks and evaluate on the 50 000 validation images. Similarly, we use the 100 000 training images from the Tiny ImageNet dataset for training and report the results on the 10 000 validation images.

### A.3 DATA AUGMENTATION

We normalize the RGB values of the images for all datasets. For CIFAR-10 and CIFAR-100, we apply random shifts by up to 16 and 8 pixels, respectively, in addition to random horizontal flips. For Tiny ImageNet, we randomly rescale the images by a factor between 0.8 and 1.0 as well as changing the aspect ratio by a factor between 0.75 and 1.33 (both uniformly distributed in the given range) and randomly crop the image to the original size. For ImageNet (Sec. B.4), we first rescale the images such that the smaller dimension measures 256 pixels. Before randomly cropping $224 \times 224$ pixel patches from the images, we apply random rescaling by a factor between 0.1 and 1.2 and changes to the aspect ratio by a factor between 0.75 and 1.33. During evaluation, a $224 \times 224$ pixel patch is cropped from the center of the image.

### A.4 TRAINING SETUP

For the experiments on CIFAR-10, CIFAR-100, and Tiny ImageNet, we use a single Nvidia GTX 1080 Ti GPU. Our method results in a minimal computational overhead. On CIFAR-10, for example, the training time per epoch of 38 seconds for the deterministic baseline is increased to 40 seconds for the NC-VIBN. We use four Nvidia GTX TITAN X for the ImageNet experiments in Sec. B.4. The last layer setup adds no computational overhead, resulting in a training tine of 56 minutes per epoch for the deterministic baseline and the NC-VIBN.

Table 5: *Overview of additional hyperparameters introduced by different methods.*

| Method | Required hyperparameters |
|---|---|
| MC Dropout | Dropout rate $p$ |
| Label-smoothing | Label-smoothing rate $\alpha$ |
| Output regularization | Weight of the regularization term $\beta$ |
| Variational approximations of the IBN objective | Lagrange multiplier $\beta$ |
| Noise-contrastive entropy regularization | No additional hyperparameters required |
| Decoder variational inference | Prior precision $\tau$ and initial dampening factor $\rho$ |

## A.5 ARCHITECTURAL DIFFERENCES

On ImageNet (see Sec. B.4), we use the original ResNet50 architecture (He et al., 2016), i.e. the first convolutional layer has stride 2 and max-pooling is applied to its output. For all other datasets, we use architectures based on ResNet18. Due to the differing image resolutions, the initial feature extraction layer of the ResNets varies for these datasets. The stride of the first layer is reduced to 1 for Tiny ImageNet, where the input resolution is $64 \times 64$ pixels. For CIFAR-10 and CIFAR-100, where the input resolution is $32 \times 32$ pixels, the first convolutional layer has stride 1 and no max-pooling is applied.

## A.6 HYPERPARAMETER SETTINGS

We use SGD with Nesterov momentum with a momentum parameter of 0.9 for all experiments. Further, we apply gradient clipping if the $\infty$-norm of the gradient is greater than 0.1. We tune the optimizer hyperparameters on the deterministic baseline and apply the same settings to all other models, only adjusting the *model* hyperparameters *specific to them* (see below). $L_2$-regularization with a weight of $10^{-4}$ is applied to the network's parameters for all experiments. For all information bottleneck and noise-contrastive information bottleneck models, the additional loss terms are scaled up linearly starting from $0.0$ to $1.0\times$ (i.e. their reported value, see below) until the first learning rate drop. For the variational information bottleneck networks with learnable Gaussian mixture priors, we set the number of mixture components to 32, sample the initial means from a centered Gaussian with variance 0.01 and initialize the variance of the mixture distributions to 1. This choice for the number of mixture components leads to a consistent improvement in negative log-likelihood for the VIBN models across all datasets. We tried to increase the number of mixture components further, but were not able to gain any additional benefits.

To select the hyperparameter $\beta$ for VIBN and NC-VIBN as well as for the baseline models with latent priors, we use a simple search over $\beta \in \{10^{-1}, 10^{-2}, 10^{-3}, 10^{-4}, 10^{-5}, 10^{-6}\}$, choosing the value with the highest validation accuracy, respectively. If two $\beta$ values have similar accuracy, the one with lower validation NLLH is preferred. The $\beta$ found for NC-VIBN tends to be lower than that for VIBN as the noise-contrastive entropy loss term offers an additional source of regularization. For learning the weight uncertainties of the decoder via variational inference, we examine possible prior precision values $\tau \in \{10^{-1}, 10^{0}, 10^{1}, 10^{2}\}$ with an initial dampening factor $\rho = 1$, see Sec. 5.4, and scale $\rho$ down by $10^{-1}$ if training is not stable during the first few iterations. We tune $\tau$ and $\rho$ after fixing the Lagrange multiplier $\beta$ to the value found as described above, because the effect of the stochastic decoder settings is less than that of the weighting of the latent KL-term. Since over-regularization for many of these settings becomes apparent during the first few training epochs, effectively only a small subset of these hyperparameter settings has to be validated.

To similarly tune the remaining baseline methods we compare to, we search for the weight of the regularization term for Dirichlet Belief Matching (Joo et al., 2020) and logit $L_2$-regularization (Joo & Chung, 2020) in $\{10^{-1}, 10^{-2}, 10^{-3}, 10^{-4}, 10^{-5}, 10^{-6}\}$, for the dropout rate (Gal & Ghahramani, 2016) and entropy regularization weight (Pereyra et al., 2017) in $\{0.05, 0.1, 0.2, 0.5\}$, and for label smoothing (Szegedy et al., 2016) in $\{0.01, 0.02, 0.05, 0.1\}$.

An overview summarizing which hyperparameters are required by different methods can be found in Tab. 5. Our NC-VIBN approach uses a variational approximation of the information bottleneck objective, noise-contrastive entropy regularization, and decoder variational inference. We detail the hyperparameters determined this way in the following sections.

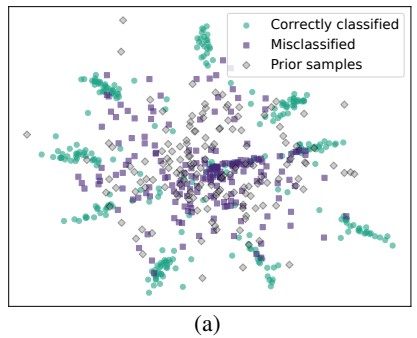
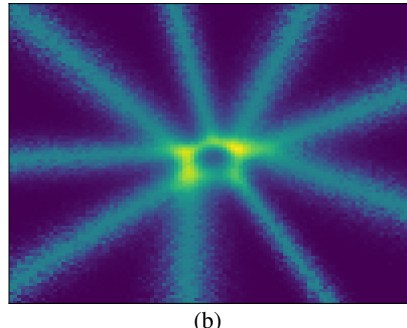

(a)                                          (b)

Figure 4: *Embeddings and samples from the latent prior (a) and assigned predictive entropy (b) for a VIBN model* with a two-dimensional information bottleneck trained on CIFAR-10. Darker areas correspond to lower entropy.

**CIFAR-10.** The networks are trained for 200 epochs with a batch size of 128. The learning rate of the optimizer is set to 0.1 and reduced by a factor of 0.2 at epochs 60, 120, and 160. For MC dropout, the parameter search yielded a suitable dropout rate of 0.2. The Lagrange multiplier $\beta$ scaling the latent KL-divergence term was identified (see above) as $10^{-3}$ for VIBN (Alemi et al., 2017), IDO (Achille & Soatto, 2018), and NC-IDO, while the search yielded $10^{-4}$ for NC-VIBN. The prior's precision $\tau$ for the decoder weights was determined as $10^{-1}$ for NC-VIBN and NC-IDO. For dampening, the initial variance is scaled down by a dampening factor of $\rho = 10^{-2}$.
In the learnable mixture prior experiments, the parameter search yielded $\beta = 10^{-3}$ for the VIBN-MP and $\beta = 10^{-4}$, $\tau = 1$, and $\rho = 10^{-1}$ for the NC-VIBN-MP network.
In the baseline experiments with output regularization methods, the parameter search yielded a label smoothing of 0.02 (Szegedy et al., 2016) and scales for the regularization term of Entropy Regularization (Pereyra et al., 2017) given as 0.2, for Dirichlet Belief Matching (Joo et al., 2020) as $10^{-4}$, and for $L_2$-regularization of the logits (Joo & Chung, 2020) as $10^{-3}$, respectively. For DUE, we tried to vary the number of inducing points to see if we can improve the in-distribution uncertainty estimation performance, and found that in the set $\{10, 20, 50, 100\}$ the models using 10 inducing points performed best.

**CIFAR-100.** For CIFAR-100, the parameter search yielded the same hyperparameter settings as for CIFAR-10.

**Tiny ImageNet.** We train the models for 60 epochs with a batch size of 64, reducing the initial learning rate of 0.1 by a factor of 0.1 at epochs 30, 40, and 50. For the MC Dropout baseline model, the parameter search yielded a dropout rate of 0.1. The weight $\beta$ of the latent KL-divergence term was identified as $10^{-3}$ for VIBN and $10^{-6}$ for NC-VIBN, while the prior precision $\tau$ of the decoder weights was determined by the search as $10^2$. The initial variance dampening parameter $\rho$ was identified as $10^{-1}$. For IDO we found $\beta = 10^{-3}$. Similarly, for NC-IDO we found $\beta = 10^{-3}$, $\tau = 10^1$, and $\rho = 10^{-1}$
For the mixture prior experiments, we determined parameters $\beta = 10^{-3}$ for VIBN-MP and $\beta = 10^{-5}$, $\tau = 1$, and $\rho = 10^{-1}$ for NC-VIBN-MP.

## B  ADDITIONAL RESULTS

### B.1  LATENT SPACE OF A VIBN MODEL

Similarly to Fig. 2, we show the predicted embeddings and assigned predictive entropy of a VIBN network trained on CIFAR-10 with a two-dimensional bottleneck in Fig. 4. While the embeddings of correctly classified data points still form clusters, which can be separated from the embeddings of incorrect examples that are closer to the latent prior, this effect is far less pronounced compared to our NC-VIBN network. Further, without the noise-contrastive loss the high-entropy regions are much more condensed, and many embeddings of incorrectly classified examples fall within regions with lower entropy, leading to poor separability .

Table 6: *Additional results for a modified last layer setup on CIFAR-10, CIFAR-100, Tiny ImageNet, and ImageNet.*

|  |  | MCR | NLLH | ECE | AUMRC | MR5% | MR10% | MR25% | MC Corr. | MC Incorr. |
|---|---|---|---|---|---|---|---|---|---|---|
| CIFAR-10 | VIBN | 5.12 | 0.203 | 2.71 | 1.08 | 2.85 | 1.61 | 0.83 | 98.50 | 80.27 |
|  | NC-VIBN *(ours)* | 4.94 | 0.176 | 1.59 | 0.76 | 2.69 | 1.42 | 0.65 | 97.63 | 76.56 |
| CIFAR-100 | VIBN | 26.01 | 1.155 | 8.71 | 7.92 | 23.44 | 20.95 | 13.60 | 90.96 | 58.72 |
|  | NC-VIBN *(ours)* | 24.26 | 1.098 | 5.92 | 7.30 | 21.50 | 28.97 | 12.28 | 87.00 | 46.44 |
| Tiny ImageNet | VIBN | 43.41 | 2.031 | 11.73 | 18.10 | 41.00 | 38.57 | 31.07 | 83.26 | 48.86 |
|  | NC-VIBN *(ours)* | 42.74 | 1.910 | 9.99 | 18.03 | 40.38 | 38.11 | 30.73 | 82.56 | 46.78 |
| ImageNet | Deterministic | 24.35 | 0.971 | 3.74 | 7.38 | 21.53 | 19.08 | 12.50 | 87.48 | 58.31 |
|  | NC-VIBN *(ours)* | 24.34 | 0.974 | 2.28 | 7.56 | 21.59 | 19.22 | 12.87 | 86.13 | 51.79 |

Table 7: *Ensembling results on CIFAR-10, CIFAR-100, and Tiny ImageNet.*

|  |  | MCR | NLLH | ECE | AUMRC | MR5% | MR10% | MR25% | MC Corr. | MC Incorr. |
|---|---|---|---|---|---|---|---|---|---|---|
| CIFAR-10 | Deterministic | 5.37 | 0.237 | 3.22 | 0.53 | 3.11 | 1.83 | 0.33 | 98.57 | 82.11 |
|  | NC-VIBN | 4.93 | 0.165 | 1.28 | 0.57 | 2.83 | 1.52 | 0.45 | 97.48 | 71.99 |
|  | Ensemble | 4.10 | 0.141 | **0.76** | 0.33 | 2.06 | 1.03 | 0.59 | 97.18 | 67.75 |
|  | NC-VIBN Ensemble | **3.73** | **0.125** | 1.32 | **0.29** | **2.05** | **0.79** | **0.15** | 96.17 | 63.37 |
| CIFAR-100 | Deterministic | 26.22 | 1.146 | 10.92 | 7.73 | 23.32 | 20.73 | 13.40 | 92.28 | 63.39 |
|  | NC-VIBN | 26.19 | 1.310 | 8.50 | 8.29 | 23.62 | 21.17 | 14.10 | 90.24 | 59.50 |
|  | Ensemble | 22.29 | 0.850 | **1.96** | **6.00** | 19.28 | 16.75 | **10.42** | 87.19 | 50.76 |
|  | NC-VIBN Ensemble | **21.31** | **0.793** | 4.56 | 6.06 | **18.66** | **16.37** | 10.76 | 83.08 | 43.49 |
| Tiny ImageNet | Deterministic | 43.09 | 2.184 | 18.26 | 17.93 | 40.64 | 38.20 | 30.63 | 87.64 | 58.70 |
|  | NC-VIBN | 41.82 | 1.756 | 6.60 | 17.57 | 39.45 | 37.15 | 29.85 | 79.61 | 44.01 |
|  | Ensemble | 37.93 | 1.675 | 4.07 | **14.38** | 35.26 | 32.58 | **24.81** | 79.65 | 43.99 |
|  | NC-VIBN Ensemble | **37.32** | **1.517** | 3.39 | 14.48 | **34.76** | **32.26** | 24.96 | 73.76 | 35.99 |

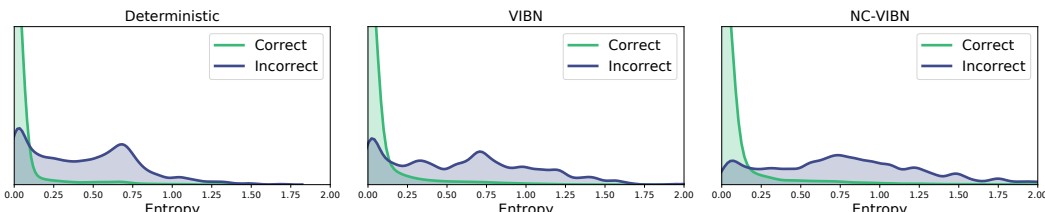

Figure 5: *Relative densities of the empirical predictive entropy distribution of correct and incorrect predictions for a deterministic, VIBN, and NC-VIBN network on CIFAR-10.*

## B.2 EMPIRICAL DISTRIBUTION OF PREDICTIVE ENTROPY VALUES

We show the empirical distribution deterministic, VIBN, and NC-VIBN networks on CIFAR-10 in Fig. 5. For all networks, the predictive entropy of correct predictions is close to zero. The distribution of the predictive entropy of the misclassifications exhibits also a peak near zero for the deterministic and VIBN network. This peak is far less pronounced for our NC-VIBN network, showing a more uniform distribution of predictive entropies for incorrect classifications. The NC-VIBN network is therefore less likely to assign low entropy to incorrect predictions.

## B.3 INFLUENCE OF THE LAGRANGE MULTIPLIER $\beta$

To study the impact of the Lagrange multiplier $\beta$ controlling the weight of the latent KL-divergence term on the NC-VIBN models, we vary $\beta$ over a range from $5 \cdot 10^{-4}$ to $2 \cdot 10^{-5}$, see Tab 8. For all choices of $\beta$, NC-VIBN performs better than VIBN (cf. Tab. 1) with respect to all evaluation metrics, showing that the improvements achieved by our NC-VIBN are robust regarding the choice of $\beta$. Further, we can observe a trade-off between the ECE and AUMRC. Increasing $\beta$ to $2 \cdot 10^{-4}$ reduces the calibration error but increases the AUMRC

Table 8: *Influence of the Lagrange multiplier $\beta$ for our full NC-VIBN model from the main paper trained on CIFAR-10.*

| $\beta$ | $5 \cdot 10^{-4}$ | $2 \cdot 10^{-4}$ | $1 \cdot 10^{-4}$ | $5 \cdot 10^{-5}$ | $2 \cdot 10^{-5}$ |
|---|---|---|---|---|---|
| MCR | 4.90 | 4.89 | 4.93 | 5.02 | 5.03 |
| NLLH | 0.172 | 0.173 | 0.165 | 0.167 | 0.170 |
| ECE | 1.43 | 0.74 | 1.28 | 1.64 | 1.89 |
| AUMRC | 0.62 | 0.59 | 0.57 | 0.55 | 0.56 |

and NLLH. Setting the value of $\beta$ even higher to $5 \cdot 10^{-4}$ overregularizes the confidence of the model and results in a higher ECE as well as NLLH and AUMRC. On the other hand, decreasing $\beta$ to $5 \cdot 10^{-5}$ slightly decreases the AUMRC while increasing the ECE.

### B.4 LAST-LAYER NC-VIBN

We additionally report results for a modified setup, where the information bottleneck is placed before the last layer and the variance of the latent embeddings is fixed, see Tab. 6. This simplification allows us to easily scale our approach to large-scale tasks, here ImageNet object classification.

**Hyperparameter settings.** For CIFAR-10, the hyperparameter search (as above) for this additional last-layer setup identified a suitable $\beta$ as $10^{-4}$ for VIBN and $10^{-6}$ for NC-VIBN and found $10^{-1}$ suitable for the decoder prior precision $\tau$. On CIFAR-100, the $\beta$ parameter was found as $10^{-4}$ for VIBN and the prior precision for NC-VIBN is changed to $10^{-2}$. For TinyImageNet, $\beta$ was found as $10^{-4}$ and the NC-VIBN decoder's prior precision as $10^{1}$.

**ImageNet.** The ImageNet models for this additional experiment are trained for 90 epochs with a batch size of 256 and a learning rate of 0.1, which is reduced by a factor of 0.1 at epochs 30, 60, and 80. The last layer NC-VIBN model scales the latent KL-divergence term with $\beta = 10^{-5}$, applies a weight prior with precision $10^{1}$ to the decoder's weights, and dampens the initial variance by a factor of $10^{-3}$. To account for the higher sampling variance caused by the larger latent space dimension, samples are rescaled by a factor of 0.75.

**Results.** The last layer setup of NC-VIBN matches the deterministic baseline on ImageNet regarding MCR, ECE, and AUMRC, while substantially improving the calibration by reducing the ECE from 0.037 to 0.023. For CIFAR-10, CIFAR-100, and Tiny ImageNet, we test both the baseline last-layer VIBN as well as our proposed noise-contrastive VIBN. For this setup again, our NC-VIBN models consistently perform significantly better than the respective VIBN models on all datasets with respect to all evaluation metrics.

Comparing this additional last-layer setup to the full setup in the main paper, see Tab. 1, we find that the last-layer setup tends to be already competitive in terms of accuracy, but that our full setup has clear benefits in terms of uncertainty estimation, where it performs strictly better regarding the ECE and NLLH. The benefit of the last-layer setup is that it easily scales to large-scale datasets such as ImageNet.

### B.5 ENSEMBLING

We compare our NC-VIBN models with deep ensembles (Lakshminarayanan et al., 2017) on CIFAR-10, CIFAR-100, and Tiny ImageNet, see Tab. 7. While an ensemble of four deterministic networks performs better than a *single* NC-VIBN network, it constitutes an almost fourfold increase of parameters and computation. A more appropriate comparison in terms of computational effort and memory requirements is thus to compare an ensemble of deterministic networks to an ensemble of NC-VIBN networks. Here, the NC-VIBN ensembles are significantly more accurate, achieve lower NLLH and comparable AUMRC for all datasets. However, ensembles of NC-VIBN networks are not calibrated as well as deterministic ensembles. This is because our NC-VIBN models are designed to achieve good calibration with a single model and averaging the predictions of multiple NC-VIBN models leads to underconfidence.

### REFERENCES

Olga Russakovsky, Jia Deng, Hao Su, Jonathan Krause, Sanjeev Satheesh, Sean Ma, Zhiheng Huang, Andrej Karpathy, Aditya Khosla, Michael Bernstein, Alexander C. Berg, and Li Fei-Fei. ImageNet Large Scale Visual Recognition Challenge. *International Journal of Computer Vision (IJCV)*, 115 (3):211–252, 2015.

