# OpenReview forum: "Noise-Contrastive Variational Information Bottleneck Networks"
_ICLR.cc/2022/Conference — ICLR 2022 Submitted_

### Official Review · Reviewer_dzx4 · 2021-10-22

**Correctness:** 4
**Technical Novelty And Significance:** 3
**Empirical Novelty And Significance:** 3
**Recommendation:** 6
**Confidence:** 4

**Main Review:**

### Novelty

This most significant contribution is the noise contrastive loss term. This builds on the one proposed by Hafner et al. (2020), but differs by drawing samples from a prior over latents (in the information bottleneck), rather than from a distribution over inputs. This is significant and, I imagine, necessary for scaling the noise-contrastive loss to high-dimensional inputs. The L2-normalisation step is an additional minor contribution. While the use of a variational distribution over neural net weights is not novel in itself, its combination with the other components is shown to be helpful.

### Strengths
- Proposed methods make sensible design choices.
- Thorough experiments with ablation studies showing the usefulness of each component.
- Good performance from the proposed methods, especially in terms of the negative log-likelihood (NLLH).

### Weaknesses/questions
- A standard technique for improving generalisation and reducing overconfidence is early stopping (e.g. by saving multiple checkpoints throughout training and taking the one which is best according to some validation metric). As far as I can tell, the authors did not do this for any baselines (or their models). I would imagine that doing so might significantly improve the baselines, particularly the deterministic classifier. The choice of which metric to use to decide when to stop (e.g. MCR or NLLH) could also have a big impact. Could the authors comment on this?
- Is the "test-time normalisation" step also performed at training time? If so, it looks very similar to Instance Normalisation (https://arxiv.org/abs/1607.08022, https://pytorch.org/docs/stable/generated/torch.nn.InstanceNorm1d.html) except that the mean is not subtracted during normalisation. Readers might find it helpful if the paper comments on the similarity between these methods.
- Can the authors give guidance on (how to choose) which of NC-IDO, NC-VIBN or NC-VIBN-MP to use in practice?


**Summary Of The Paper:**

This paper attempts to improve uncertainty estimates for neural network-based classification. To do so, they combine various techniques: the information bottleneck, a variational distribution over neural net weights, a noise contrastive loss term, and an L2-normalisation step for the last part of encoder. They present experimental results justifying these choices (including, notably, significantly improved NLL compared to the baselines).

**Summary Of The Review:**

Overall, I am giving a score of 6. The novelty is not ground-breaking but the authors achieve good results for an important topic. Importantly, the design choices made are well supported by ablation studies.

---

### Official Review · Reviewer_mHTV · 2021-11-01

**Correctness:** 2
**Technical Novelty And Significance:** 2
**Empirical Novelty And Significance:** 2
**Recommendation:** 3
**Confidence:** 3

**Main Review:**

Strengths
* The paper is generally easy to read.
* The empirical comparison with baselines is rather comprehensive which includes different metrics. The proposed method leads to consistent better performance over output regularization methods.

Weakness
* The proposed method consists of three components: 1) Bayesian learning for classifier weight; 2) noise-contrastive loss; 3) test-time normalization. But 1) & 3) are not very well motivated and disconnected with the previous empirical analysis, and they are ablation-studied separately in Table 3. It could be better to reorder Sec. 5 by introducing 2) first (which is closely related to previous analysis) and then 1) & 3) with more motivating explanations (especially for 3) that seems to be hand-wavy). Furthermore, from Table 3, it seems that 1) & 3) indeed improve both the accuracy and calibration a lot, and 2) improves calibration but slightly hurts the accuracy (MCR). More detailed discussion about the effectiveness of each component needs to be included.
* The interplay between variational information bottleneck (VIB) loss and noise-contrastive loss is not clear. The VIB loss encourages the encoder latent distribution to be close to the latent prior (by a KL divergence), but the noise-contrastive loss encourages high predictive entropy for the latent prior. Would it also encourage the encoded latents to have a high entropy, and why a separation between correctly classified latents and the prior latents would happen from a theoretical perspective? It would be great to include a more detailed discussion.
* The claim that the noise-contrastive loss increases the separability of correct and incorrect samples is not supported by enough empirical evidence. From Table 1, it is hard to tell if VC-VIBN consistently improves the relevant metrics (including AUMRC, MRX%, MC Corr., MC Incorr.) over its VC-VIBN counterpart and the MC Dropout baseline, especially on large datasets like CIFAR-100 and Tiny ImageNet.

Additional questions & comments
* It seems that the proposed loss only regularizes the predictive entropy of noise samples (from the latent prior) but no contrastive loss used, why is it called the noise-**contrastive** loss?
* Since Fig. 2(a) is plotted with NC-VIBN, it is not convincing enough that the misclassified samples would get clustered with latent prior samples for the original VIBN model without the NC loss, similarly to a chicken-and-egg problem. It would be great to have a plot with the original VIBN model (i.e., Fig. 4) in the main body. A discussion for the comparison between Fig. 2 and Fig. 4 can also highlight the effect of the noise-contrastive loss.
* Typo in Eq. (4) and above: $p(y_n | z_n, \psi)$ -> $p(y_n | x_n, \psi)$, $\mathcal{D}(z_n, y_n)$ -> $\mathcal{D}(x_n, y_n)$
Since the empirical comparison involves several metrics, it would be more friendly for the reader to indicate for each metric whether it is lower is better or higher is better in the tables.


**Summary Of The Paper:**

This paper first empirically observes that output regularization methods for reducing model overconfidence improves calibration but make correct and incorrect predictions indistinguishable based on predictive uncertainty.  To tackle the issue, it proposes a noise-contrastive loss and some architectural refinements based on the variational information bottleneck. Empirical results demonstrate both improved accuracy and calibration over output regularization methods.

**Summary Of The Review:**

The central proposal of the paper is the noise-contrastive loss for VIBN, but its effectiveness is obfuscated by other two proposed tricks that are not well motivated and studied. Though the proposed combined method leads to better accuracy and calibration than output regularization methods, the claim that the noise-contrastive loss increases the separability of correct and incorrect samples is not clearly backuped with sufficient empirical evidence. Thus, I recommend a ‘reject’ for the paper but will consider raising my score if my concerns are well addressed.

---

### Official Review · Reviewer_DCEH · 2021-11-02

**Correctness:** 3
**Technical Novelty And Significance:** 3
**Empirical Novelty And Significance:** 3
**Recommendation:** 5
**Confidence:** 3

**Main Review:**

Please see my detailed comments below:

W1: Is there any intuition why “embeddings of hard examples tend to be aligned with the latent prior” and “is easier to be optimized” Is there any empirical evidence or reference supporting such argument? (Description and motivation above Eq. (6). It seems this is the motivation of the paper, and I think it is better to provide some empirical evidence of such claim.

W2: I am not sure if the "noise-contrastive" loss is a proper description of the penalty Eq. (6), since it seems to be an entropy term in absence of any contrastive flavor. Please could you clarify what is what is the connection between the entropy term with the “noise-contrastive”.

W3: Novelty. It seems to be that the only difference between the proposed framework and conventional VAE framework is the entropy term and the additional prior on the network weights. I think there have been many works investigating the entropy term in similar conditional generative process context though. Take for example, in paper [a] [b], the entropy loss has already been explored thoroughly in order to encorage feature seperability, when it is associated with a generative loss, where a locally-Lipschitz constraint is even further imposed to guarantee the correctness of such entropy loss in [a]. It seems to me that the minor difference here is that the generative model now belongs to a variational inference family such as VAE rather than adversarial training (i.e., GAN) in [a] [b]. Please correct me if I misunderstood anything.

W4: Empirical evidence. The paper claims that harder samples shall benefit from the proposed procedure owing to the additional entropy constraint. However, it seems to me that the empirical evidence does not fully support this claim. I noticed that on Tiny ImageNet100 which is seemingly more complex than CIFAR10/100, the proposed method could hardly beat the other state-of-art method, leaving me doubt the effectiveness of the method.

If possible, please clarify the above concerns during the rebuttal. I am happy to increase my score if the above issues or misunderstandings are properly addressed.


[a] Shu et al., “A DIRT-T Approach To Unsupervised Domain Adaptation” ICLR 2018.

[b] Miyato et al., “Virtual Adversarial Training: A Regularization Method for Supervised and Semi-Supervised Learning”., IEEE Transactions on Pattern Analysis and Machine Intelligence 2019


**Summary Of The Paper:**

The paper proposes a new model NC-VIBN for the classification tasks. To ensure the separability between classes, the authors introduced an entropy term into the conventional VAE framework.

**Summary Of The Review:**

I cannot recommand the acceptance of the paper until the above issues and concerns are addressed. But I would be happy to increase my score if any potential misunderstandings are clarified. Please see detailed comments above regarding technical concerns.

---

### Official Review · Reviewer_r8pu · 2021-11-02

**Correctness:** 4
**Technical Novelty And Significance:** 3
**Empirical Novelty And Significance:** 2
**Recommendation:** 5
**Confidence:** 3

**Details Of Ethics Concerns:**

There is no ethical concern in this paper.

**Main Review:**

**Pros**
The authors provide an intriguing solution to a problem concerning the uncertainty estimation of machine learning models. Although the proposed method is limited to the variational bottleneck network, it is simple and practical and empirically proves that it is more effective than VIBN and IDO in the separability problem.

**Cons**
My main concern is empirical verification. The reviewer thinks that the proposed method should be tested to determine if it works well with different convolutional networks other than ResNet. Also, the reviewer wonders the proposed method will work if we apply distribution matching methods in input space, such as augmentation strategies [1, 2], to the VIBN. In the appendix, the authors are experimenting with relatively simple augmentation policies only. It is crucial to clarify how the proposed loss term works if strong augmentation policies are used.

[1] AutoAugment: Learning Augmentation Policies from Data, Cubuk et al., *CVPR* (2019)

[2] Fast AutoAugment, Lim et al., *NeurIPS* (2019)

**Summary Of The Paper:**

This work suggests a *noise-contrastive loss* for variational information bottleneck networks to resolve the poor performance at separating correct and incorrect predictions in regularization methods. Standard regularization methods suffer *separability problems*: models indiscriminately regularize confidence to improve calibration perform much worse at differentiating correct from wrong predictions. The authors address this problem in the variational information bottleneck, suggesting a remedy that utilizes the distribution matching in the latent space. The authors have applied the proposed loss term to the two variational approximations of the information bottleneck, the deep variational information bottleneck (VIBN, Alemi et al., 2017) and information dropout (IDO, Achille & Soatto, 2018).

**Summary Of The Review:**

The paper is generally well-written and given the technical content. The reviewer considers the proposed method quite interesting, but some issues are to be addressed empirically.

---

### Decision · Program_Chairs · 2022-01-20

**Decision:**

Reject

**Comment:**

The paper proposes a classification method that improves model calibration using variational information bottlenecks and a noise-contrastive loss.

Unfortunately, the authors' were not able to participate in the discussion of the paper with the reviewers. Given this, the reviewers raised several unaddressed concerns: First, it was argued that the different components of the proposed method required additional justification, in particular with regards to novelty. Second, reviewers argued that the paper required additional empirical validation, for example by testing if it works well with different convolutional methods such as ResNets.

Given these concerns, a consensus was reached that this paper should be rejected which is also my recommendation,